

# Development of a Japanese version of the Psychological Ownership Scale

Sayo Iseki[1], Kyoshiro Sasaki[2] and Shinji Kitagami[3]

[1] Department of Management, Chukyo University, Nagoya, Aichi, Japan
[2] Faculty of Informatics, Kansai University, Takatsuki, Osaka, Japan
[3] Department of Psychology, Graduate School of Informatics, Nagoya University, Nagoya, Japan

## ABSTRACT

The present study addresses the need for a valid instrument for measuring dimensions of psychological ownership, including that of owned and non-owned objects, for use in the language and culture of Japan. Although the theory of psychological ownership has expanded self-extension theory, the most widely used scale of psychological ownership does not measure the extent to which one feels that it (the owned object) is a part of them. Thus, the present study aimed to develop a Japanese version of the Psychological Ownership Scale (POS-J) and examine its reliability and validity. Study 1 measured the POS-J of an owned object, finding the POS-J to have a two-factor structure (possession-self link and feeling of ownership) and its internal consistency and reliability to be adequate. Moreover, POS-J scores were positively correlated with perceived control and self-extension tendency, but not monetary value, indicating that conceptual validity was generally supported. To confirm whether the POS-J could be used for a non-owned object, Study 2 rephrased the expressions of item descriptions and examined the effect of imagining touching a non-owned object on the POS-J scores, showing that doing so increased the POS-J scores for the object. Our findings suggest that the POS-J is a reliable and valid measure of the psychological ownership of owned and non-owned objects for use in Japan.

Corresponding author
Sayo Iseki,
s-iseki@mecl.chukyo-u.ac.jp

## INTRODUCTION

When it comes to our possessions, we have a sense that they "belong to us," meaning we also have psychological ownership over them (*Pierce, Kostova & Dirks, 2001*, *2003*). However, we also have psychological ownership over things we do not actually own (*e.g.*, *Reb & Connolly, 2007*). For example, a person is likely to have strong psychological ownership over the desk they use at work, even though, in reality, it is owned by their employer. In other words, psychological ownership is distinct from legal ownership. Legal ownership is recognized by society and is upheld by law while psychological ownership is perceived, regardless of actual legal ownership.

Previous research has addressed the significance of object ownership and the psychological effects of ownership (*e.g.*, *Beggan, 1991*; *Dittmar, 1992*; *Furby, 1978a*, *1978b*). Later studies focused on the feeling that something "is mine" (*i.e.*, psychological

ownership) that arises regardless of legal ownership, especially in the field of organizational psychology. For example, employees who work for a company often use the expression "*my* company," suggesting that they feel the company's psychological ownership despite having no legal ownership. Psychological ownership of an organization has been shown to promote commitment to the organization and extra-role behavior, increase productivity, and enhance job satisfaction (*Pierce & Rodgers, 2004*; *Vandewalle, Van Dyne & Kostova, 1995*; *Van Dyne & Pierce, 2004*). In addition to several studies in organizational psychology, psychological ownership has also been investigated from the perspectives of economics and consumer psychology.

Another subject closely related to the emergence of psychological ownership is the endowment effect, which causes people to overvalue objects they own (*Kahneman, Knetsch & Thaler, 1990*). Specifically, the amount of money people are willing to pay to obtain an object (willingness to pay) is lower than the amount of money they are willing to accept to give away the object after gaining ownership (willingness to accept), and the effect increases as psychological ownership increases (*Brasel & Gips, 2014*; *Shu & Peck, 2011*). Thus, psychological ownership is key for causing the endowment effect, and these are often addressed in the same context.

A similar concept, the mere ownership effect, creates a bias in which people feel that a self-owned object is more appealing than a non-owned object (*Beggan, 1992*). As soon as objects are recognized as owned by oneself, they are evaluated more positively than those owned by others (*Kim & Johnson, 2012*). There are two primary explanations for why the mere ownership effect occurs (*Morewedge & Giblin, 2015*). First, when a connection forms between the self and an object, the object is incorporated into the individual's self-concept, thereby becoming a part of the self, and is then attributed traits associated with the self (*Belk, 1988*; *Weiss & Johar, 2013*). Thus, the more positively individuals evaluate themselves, the more positively they will evaluate their owned objects (*Gawronski, Bodenhausen & Becker, 2007*). Second, mere ownership effect occurs due to the self-reference effect: people more easily recall self-relevant things because self-referent processing contributes to memorization (*Symons & Johnson, 1997*). Moreover, the ease of access to information about owned objects leads people to evaluate them more positively (*Carmon & Ariely, 2000*). Accordingly, both the abovementioned endowment effect and the mere ownership effect indicate that self-owned objects tend to be evaluated as superior to other-owned objects, and both effects are widely accepted as arising due to a strong link between "self" and "object" (*Beggan, 1992*; *Belk, 1988*; *Furby, 1978b*; *Pierce, Kostova & Dirks, 2003*). It has also been shown that the superior evaluation of owned (*vs.* others') objects is modulated by the evaluation of oneself and by cultural factors (*Maddux et al., 2010*). In other words, how one favorably evaluates the object that they own (or perceive ownership over) is dependent on how they evaluate themselves as well as their cultural context.

Such a connection between self and object has also been found in studies using implicit indicators. For example, using an implicit association test, *LeBarr & Shedden (2017)* found that reaction times were faster when self-related words required pressing the same response key as a self-owned object. These effects have even been confirmed immediately

after an object becomes self-owned, indicating the connection between self and object is formed in a matter of minutes. *Turk et al. (2011)* examined cognitive processing at the moment when self-ownership was established and found that the amplitude of the P300 event-related potential component increases in response to cues indicating that an object is self-owned. This may be because owned objects are highly self-relevant and thus draw selective attention. These behavioral and physiological studies suggest that the connection between self and object is involved in implicit/automatic cognitive processing.

Furthermore, the tendency to estimate a high value for self-owned objects and evaluate them as preferable has been observed even when there is no legal ownership. For example, *Peck & Shu (2009)* found that simply touching an object that does not belong to oneself increases psychological ownership and results in a higher valuation. This is because touching that creates a connection between the self and the object improves perceived control and fosters psychological ownership. Additionally, *Asatryan & Oh (2008)* reported that consumers perceive restaurants they regularly visit to be "their restaurants." Moreover, the more elaborately one imagines a product before purchase, the more likely psychological ownership will increase (*Kamleitner & Feuchtl, 2015*), and the emergence of psychological ownership has been shown to increase attachment (*Kamleitner & Feuchtl, 2015*), user's intention toward subscription services (*Danckwerts & Kenning, 2019*), user participation in social media (*Kwon, 2020*), prosocial behavior (*Jami et al., 2021*), and willingness to pay for an extended warranty (*Lessard-Bonaventure & Chebat, 2015*). These findings suggest that the emergence of psychological ownership impacts purchase decisions because of their increased value in the buyer's mind.

## Extended self

When measuring the psychological ownership of a given object, most previous research has used scales that directly ask to what extent one feels they are the owner of an object based on *Pierce, Kostova & Dirks (2001)* (*e.g.*, *Kamleitner & Feuchtl, 2015*; *Menard, Warkentin & Lowry, 2018*; *Peck, Barger & Webb, 2013*; *Peck & Shu, 2009*; *Shu & Peck, 2011*; *Van Dyne & Pierce, 2004*). However, such scales may grasp only one aspect of psychological ownership. This is because psychological ownership reflects not only cognitive aspects, such as awareness of or thoughts about the object but also emotional aspects such as personal significance (*Pierce, Kostova & Dirks, 2001*, *2003*). In other words, psychological ownership is closely linked to one's self-concept (*Hillenbrand & Money, 2015*). Given the suggestion that the final stage of psychological ownership is an integration of the object and self (*Pierce, Kostova & Dirks, 2001*, *2003*), psychological ownership is thought to reflect a mental state in which the object and self are closely connected (*Furby, 1978a*, *1978b*; *Wilpert, 1991*).

A "self" that which goes beyond the perception of "I" and extends to objects in the physical environment, is conceptualized as an "extended self" (*Belk, 1988*; *Dittmar, 1992*). We come to perceive an object as an extended self when we control it, gain knowledge about it, or invest in it (*Belk, 1988*). It has been suggested that the reason we feel like we have lost a part of ourselves when we lose an object that is important to us, or like there has

been an assault on our identity when we have our possessions stolen, is that we perceive our owned objects as extensions of ourselves (*e.g.*, *Duncan, 1976*).

## Purpose of this study

The purpose of this study is to develop a Japanese version of the Psychological Ownership Scale (POS-J) and to examine its reliability and validity. As mentioned above, although the theory of psychological ownership has expanded self-extension theory, the most widely used scale of psychological ownership does not measure the extent to which one feels that it is a part of them (*e.g.*, *Peck & Shu, 2009*). *Jussila et al. (2015)* also cite the need to measure psychological ownership from two perspectives: one's possession and as a part of oneself.

Therefore, this study was focused on the Psychological Ownership Scale developed by *Walasek, Matthews & Rakow (2015)*. Their scale comprises two factors: *possession-self link* and *feeling of ownership*. Possession-self link is based on items used by *Ferraro, Escalas & Bettman (2011)* and was created by combining a scale measuring the extent to which an owned object is considered an extended self (*Sivadas & Venkatesh, 1995*) and a scale measuring the connection between the self and a brand (*Escalas, 2004*). Meanwhile, the scale used to measure the feeling of ownership was created to measure the extent to which one feels that they are the owner of an object based on *Pierce, Kostova & Dirks (2001)*. In this way, *Walasek, Matthews & Rakow (2015)* Psychological Ownership Scale attempts to capture psychological ownership from two sides, both the feeling that the object is part of oneself and that it belongs to oneself.

Most previous studies on psychological ownership have focused on the latter of these two aspects, only measuring the degree of feeling of ownership (*e.g.*, *Kamleitner & Feuchtl, 2015*; *Menard, Warkentin & Lowry, 2018*; *Peck, Barger & Webb, 2013*; *Peck & Shu, 2009*; *Shu & Peck, 2011*; *Van Dyne & Pierce, 2004*). Therefore, such studies can be categorized as examinations of the effects of feelings of ownership described by *Walasek, Matthews & Rakow (2015)*.

As previously discussed, in recent years, there has been an increased effort to accurately grasp the extent of psychological ownership, which we hold toward things, in fields ranging from basic research in cognitive psychology and cognitive neuroscience to applied organizational psychology and consumer behavior studies to marketing practice. Nonetheless, the number of studies on psychological ownership in Japan is still limited (*e.g.*, *Iseki & Kitagami, 2016*; *Kwon, 2021*). One reason for this is the lack of measurement tools for Japanese speakers. The problem is not simply that reliability and validity have not been sufficiently verified, but that no Japanese scale usable for both owned and non-owned objects has been developed.

The present study aimed to develop a Japanese version of the Psychological Ownership Scale (POS-J) and examine its reliability and validity by measuring the psychological ownership of owned objects (Study 1) and non-owned objects (Study 2). Specifically, in Study 1, we developed the POS-J, a Japanese translation of *Walasek, Matthews & Rakow (2015)* scale and examined its reliability and validity. Study 1 measured the psychological ownership of objects that the participants actually owned. In Study 2, we

aimed to develop a scale that could be used without confusion when measuring the psychological ownership of objects that participants did not own (*i.e.*, no legal ownership version). To do so, we focused on the effect of "haptic imagery," the robustness of which has been confirmed in previous studies (*e.g.*, *Iseki & Kitagami, 2016*; *Peck, Barger & Webb, 2013*), to determine whether psychological ownership increases when viewing pictures of non-owned objects and imagining touching them. For Study 2, we rephrased the questions developed in Study 1 to create a version of the POS-J for non-owned objects.

## STUDY 1
### Materials and methods

In Study 1, we created the POS-J, a Japanese version of *Walasek, Matthews & Rakow (2015)* scale and administered three separate surveys with different contents. In Survey 1, we examined the reliability and validity of the scale. In Survey 2, we re-administered the survey with added indicators to examine validity. In Survey 3, we verified the scale's test–retest reliability.

#### *Ethics statement*
The present study was approved by the Ethics Committee on Human Experimentation at Nagoya University, Japan (approval number: NUPSY-200503-I-01) and Chukyo University, Japan (approval number: 2020-001). All study procedures adhered to the ethical standards of the relevant institutional committees and the Declaration of Helsinki. Only those who understood the summary of the survey shown on the recruitment page and consented to participate completed the surveys.

#### *Participants*
Detailed information about the estimation of the necessary sample size and data exclusion criteria can be found in the Supplemental Article. Participants in Survey 1 were recruited using a crowdsourcing service (Yahoo! Inc., Sunnyvale, CA, USA). A total of 1,212 individuals participated in the survey, and data from 376 participants were excluded. Ultimately, data of 836 participants between the ages of 15 and 81 (531 males, 298 females, 7 no responses; mean age 44.00 years, SD = 10.50) were analyzed (Sample 1). Participants in Survey 2 were recruited using a crowdsourcing service (CrowdWorks, Inc., Shibuya City, Tokyo, Japan). Following *Walasek, Matthews & Rakow (2015)*, in Survey 2, responses were also excluded if WTA was 10 times the market price or more. Thus, a total of 1,555 individuals participated in the survey, and data from 714 participants were excluded. Ultimately, data of 841 participants between ages 19 and 71 (355 males, 477 females, 9 no response; mean age 39.12 years, SD = 10.51) were analyzed (Sample 2). Participants in Survey 3 were recruited from university students. The survey was administered twice, with a 4-week interval between each administration. A total of 205 students participated in the first administration and 194 in the second. Participants who did not respond to either administration were excluded. Ultimately, the data of 105 participants between the ages of 18 and 21 (57 males, 48 females; mean age 18.28 years, SD = 0.59) were analyzed (Sample 3).

### Translation of the POS-J

We translated *Walasek, Matthews & Rakow (2015)* Psychological Ownership Scale into Japanese, after which we requested the back translation of it from an English editing and translation service (Cactus Communications). Next, the original authors (Walasek and Ferraro) checked the back translation for consistency with the original scale. All items of the POS-J are listed in Table S1. Each item was answered on a five-point scale from "*completely disagree*" to "*completely agree.*"

### Measurements

The following indicators were used to examine the validity of Surveys 1 and 2.

First, we measured perceived control over an object, which has been shown to induce a sense of ownership and that the object is a part of oneself (*Belk, 1988*; *Furby, 1978a*; *Peck, Barger & Webb, 2013*; *Pierce, Kostova & Dirks, 2003*). There is also a conceptual model that cites perceived control as the cause of psychological ownership (*Pierce, Kostova & Dirks, 2003*; *Jussila et al., 2015*). It has further been suggested that interaction between digital technology and the self promotes perceived control and leads to the emergence of psychological ownership (*Kirk, Swain & Gaskin, 2015*). Therefore, a positive correlation between perceived control (the subjective feeling that one is successfully controlling an object as they intend) of the object and psychological ownership is predicted. Specifically, a moderate-to-strong positive correlation between the feeling of ownership and perceived control is expected (*Atasoy & Morewedge, 2018*; *Iseki & Kitagami, 2016*; *Peck, Barger & Webb, 2013*). While controlling an object creates a link to the self, objects that are controlled by others or that one is unable to control are not seen as a part of the self (*Seligman, 1975*). Thus, it is predicted that there is also a positive correlation between possession-self link and perceived control, but that the association with the feeling of ownership will be stronger (*Walasek, Rakow & Matthews, 2017*). For these reasons, we measured perceived control over an object to examine its validity in Surveys 1 and 2. Specifically, participants responded to the item "I feel I can control that [object] how I want to" on a seven-point scale from "*completely disagree*" to "*completely agree*" to measure perceived control over the object.

Next, we measured willingness to accept (WTA) and willingness to pay (WTP) using the WTA/market price (WTA index), WTP/market price (WTP index), and WTA/WTP (endowment effect index). People have been shown to estimate a higher monetary value for things they feel psychological ownership over (*Shu & Peck, 2011*) and that the endowment effect is mediated by psychological ownership (*Brasel & Gips, 2014*; *Shu & Peck, 2011*). Thus, a moderate or stronger positive correlation between the feeling of ownership and monetary value indicators is expected (*Shu & Peck, 2011*). A weak positive correlation between possession-self link and monetary value has also been suggested (*Ferraro, Escalas & Bettman, 2011*). Accordingly, in Surveys 1 and 2, we measured the indices—WTA/market price (WTA index), WTP/market price (WTP index), and WTA/WTP (endowment effect index)—as indicators of monetary value; we also checked for associations with each of these factors to examine validity. A Japanese translation of *Walasek, Matthews & Rakow (2015)* was used to gauge perceptions of monetary value.

Specifically, participants responded to the items "How much would somebody have to pay for you to give that [object] away?" (WTA), "Suppose you lost that, and it was possible to buy it back. What would be the highest price you would be willing to pay to get it back?' (WTP), and "How much is that typically sold for?" (market price), with the amount of money (Japanese yen). As high and low WTA and WTP are affected by the original market price, we used WTA/market price as a WTA index and WTP/market price as a WTP index, as in *Walasek, Matthews & Rakow (2015)*. We also used WTA/WTP as the endowment effect index, and these indices were used in the analyses.

In Survey 2, we measured self-extension tendency (*Ferraro, Escalas & Bettman, 2011*), which differs between individuals, in addition to the above items. It has been suggested that individuals who are more likely to extend to an owned object are more likely to have higher psychological ownership (*Pierce, Kostova & Dirks, 2003*). Specifically, a moderate-to-strong positive correlation between possession-self link and self-extension tendency is predicted (*Ferraro, Escalas & Bettman, 2011*; *Walasek, Rakow & Matthews, 2017*). On the other hand, self-extension tendency does not seem to capture the feeling of ownership directly, which suggests that there will be a weak correlation between the two. Japanese translations of eight items from *Ferraro, Escalas & Bettman (2011)*, such as "I have a special bond with my favorite possessions," were used to measure self-extension tendency. Participants responded to each question using a five-point scale from "*completely disagree*" to "*completely agree*" (Table S2). Additionally, the Need for Touch Scale (*Peck & Childers, 2003*) was used as a dummy questionnaire in Survey 2; however, we did not analyze the data from the dummy questionnaire because it was irrelevant to this study.

### Procedures

Following the procedure used in Study 2 from *Walasek, Matthews & Rakow (2015)*, the surveys were conducted online. First, all participants were asked to provide their age and gender. Subsequently, participants were asked to identify a cherished possession as follows: "For the next task, we need you to think of one of your favorite and cherished material possessions. It does not matter how expensive the object is but try to pick something that costs around 30,000 yen. It should be something that is important to you and helps to define who you are. It does not necessarily have to be something that you purchased yourself. It is very important that you identify one object before proceeding with the task." In Survey 1, the monetary value was restricted to "an important object of around 30,000 yen" similar to *Walasek, Matthews & Rakow (2015)*, but this restriction was not applied in Survey 2 or Survey 3 (the reason for this is explained later). After this, participants were asked to respond to the nine items of the POS-J with respect to the object they had chosen. Items were randomly presented to each participant. Next, to examine the validity of the POS-J, participants were asked to rate their perceived control over their chosen object and provide their WTA and WTP and the market price as monetary amounts in Japanese yen.

Given that it is common to see undesirable response behavior of minimizing effort when conducting online surveys (*e.g.*, *Oppenheimer, Meyvis & Davidenko, 2009*), it is essential to

**Table 1  Outline of Study 1.**

| Survey 1 | Survey 2 | Survey 3 |
|---|---|---|
| Sample 1 ($n$ = 836) | Sample 2 ($n$ = 841) | Sample 3 ($n$ = 105) |
| Testing a factor structure | Testing a factor structure | Testing reliability |
| EFA | CFA | Test–retest method |
| CFA | Testing validity | |
| Testing validity | Perceived control | |
| Perceived control | WTA/MP | |
| WTA/MP | WTP/MP | |
| WTP/MP | WTA/WTP | |
| WTA/WTP | Self-extension tendency | |

Note:
EFA, exploratory factor analysis; CFA, confirmatory factor analysis; WTA, willingness to accept; WTP, willingness to pay; MP, market price.

exclude participants for whom analyses showed such response behaviors. Thus, we incorporated an instructional manipulation check (IMC; *Miura & Kobayashi, 2016*; *Oppenheimer, Meyvis & Davidenko, 2009*) and an attention check question (ACQ; *e.g., Aust et al., 2013*; *Oppenheimer, Meyvis & Davidenko, 2009*). The IMC was included in the phase in which participants were asked to think of a cherished possession. The dummy question "Do you like the font of these instructions?" and three responses ("*dislike,*" "*neither,*" or "*like*") were presented along with the instructions to call to mind a cherished possession. The instructions also included the direction to "select 'neither'" in answer to the question "Do you like the font of these instructions?" A numerical calculation (74–47 = ?) was also inserted within the scale as an ACQ, and participants were asked to select from among the five response options.

In Survey 2, in addition to the above procedure, participants were also asked to respond to the self-extension tendency scale and filler items at the start of the survey. Survey 3, which was administered twice (with an interval of 4 weeks), did not measure the indicators used to examine validity. The participants were also instructed to choose the same object for administration two that they picked at administration one. An outline of Study 1 is shown in Table 1.

### Data analysis

We conducted parallel analysis of the POS-J. Consistent with *Walasek, Matthews & Rakow (2015)*, we then performed a principal component analysis (oblimin) on the data acquired and used the split-half method to confirm the reliability of the results. Next, we performed an exploratory factor analysis (maximum likelihood ratio and promax[1] rotation) for half of the valid data. We also calculated the KMO and performed Bartlett's test of sphericity. Next, we performed a confirmatory factor analysis using structural equation modeling for the other half of the valid data. Exploratory factor analysis and confirmatory factor analysis were performed using different data to ensure sample independence (*Iwasa, Tanaka & Yamada, 2016*). Goodness-of-fit indices including the Comparative Fit Index, Tucker–Lewis Index, the root mean square error of approximation,

[1] Although we had originally adopted varimax rotation, we changed to promax rotation as suggested by the editor. In addition, consistent with the PCA, we expected the factors to be correlated. Therefore, we considered it reasonable to use an oblique rotation.

and the standardized root-mean residual were examined to determine how well the model fit the data. Following the existing recommendations (*e.g.*, *Schreiber et al., 2006*), indices such as CFI > 0.95, TLI > 0.95, RSMER < 0.06 to 0.08 and SRMR < 0.08 were used to as good-fit criteria.

The reliability of the POS-J was tested using Cronbach's alpha coefficient (Survey 1) and test–retest intraclass correlation coefficients (Survey 3). Prior to examining the validity, we calculated the descriptive statistics and correlation coefficients between the variables in both Surveys 1 and 2. To examine the validity of the POS-J, we performed a multiple regression analysis with POS-J subscales as the dependent variable and perceived control, WTA index, WTP index, endowment effect index as independent variables in Survey 1. In addition, self-extension tendency was added as an independent variable in Survey 2. For the criterion related validity and the construct validity, our predictions of the strength and weakness of the association between each variable are as follow. It is expected that there is also a positive correlation between possession-self link and perceived control, but that the association with the feeling of ownership will be stronger. In addition, a moderate or stronger positive correlation between the feeling of ownership and monetary value (*i.e.*, WTA index, WTP index, endowment effect index) indicators is expected. A weak positive correlation between possession-self links and monetary value has also been suggested. Furthermore, a moderate-to-strong positive correlation between possession-self link and self-extension tendency is predicted. On the other hand, there will be a weak correlation between the feeling of ownership and self-extension tendency.

## Results

In this study, all analyses were performed using the R version 4.1.2 (R Development Core Team, Vienna, Austria) and 'GPArotation,' 'lavaan,' and 'car' packages for factor analyses and regression analyses.

### Factor analysis

Parallel analysis of the POS-J using sample 1 found a two-factor structure to be valid. Next, we performed a principal component analysis (oblimin) on the data acquired and used the split-half method to confirm the reliability of the results. As a result, items were divided into two components in the same way they had been in *Walasek, Matthews & Rakow (2015)* (Component 1: principal component contribution = 3.56, explained variance = 60%; Component 2: principal component contribution = 2.37, explained variance = 40%). When the participants were divided into two groups for analysis, both Group 1 (Component 1: principal component contribution = 3.50, explained variance = 59%; Component 2: principal component contribution = 2.39, explained variance = 41%) and Group 2 (Component 1: principal component contribution = 3.61, explained variance = 60%; Component 2: principal component contribution = 2.38, explained variance = 40%) demonstrated the same structure as above.

Next, we performed an exploratory factor analysis (maximum likelihood ratio and promax rotation) for half of the valid data. We also calculated the KMO and performed Bartlett's test of sphericity. As shown in Table 2, this resulted in the identification of two

**Table 2 Exploratory factor analysis and Cronbach's alpha of POS-J.**

| Items | Factor 1 | Factor 2 |
|---|---|---|
| That is part of who I am | **0.65** | 0.05 |
| I derive some of my identity from that | **0.77** | 0.06 |
| That is central to my identity | **0.86** | −0.06 |
| That helps me narrow the gap between what I am and what I try to be | **0.57** | 0.03 |
| That helps me to achieve the identity I wish to have | **0.69** | 0.12 |
| That and I have a lot in common | **0.69** | −0.13 |
| I feel like I own that | −0.01 | **0.82** |
| I feel like that is my possession | −0.10 | **0.97** |
| I feel a very high degree of personal ownership of that | 0.09 | **0.66** |
| Eigenvalue | 4.32 | 1.57 |
| Contribution | 0.34 | 0.23 |
| α coefficient | 0.86 | 0.85 |

**Note:**
Factor loadings in bold denote meeting the criteria (>.40).

factors with items divided in the same way as in *Walasek, Matthews & Rakow (2015)* (Factor 1: contribution = 34%, explained variance = 59%; Factor 2: contribution = 23%, explained variance = 41%). We named Factor 1 (possession-self link) and Factor 2 (feeling of ownership)." Note that the KMO value was 0.86, and Bartlett's test of sphericity was significant ($\chi^2(36) = 1{,}724.95$, $p < 0.001$).

We performed a confirmatory factor analysis using structural equation modeling for the other half of the valid data. The two-factor model, found using exploratory factor analysis, was used as the model. The results demonstrated that the model had a high goodness of fit (CFI = 0.97, TLI = 0.96, RMSEA = 0.07, SRMR = 0.04). Factor loadings for confirmatory factor analysis are shown in Table S3.

Confirmatory factor analysis performed again for the same model using Sample 2 found goodness of fit to be at a satisfactory level (CFI = 0.95, TLI = 0.94, RMSEA = 0.09, SRMR = 0.05). For Sample 2 only, although the RMSEA did not meet the goodness-of-fit criterion (RMSEA < 0.06 to 0.08) of previous studies (*e.g.*, *Schreiber et al., 2006*), the CFI, TLI, and SRMR met the good level.

### Validity

When performing multiple regression analysis using Sample 1 and Sample 2, we calculated the descriptive statistics and correlation coefficients between the variables (Table 3). Next, for Survey 1, we performed multiple regression analysis with the possession-self-link score as the dependent variable and perceived control, WTA index, WTP index, and endowment effect index as independent variables. The results demonstrated that only perceived control and WTP index significantly predicted the possession-self-link score in a positive direction ($\beta = 0.21$, $p < 0.001$; $\beta = 0.07$, $p = 0.03$). Next, when the same multiple regression analysis was performed using the feeling of ownership score as the dependent variable, only perceived control significantly predicted the feeling of ownership

**Table 3 Descriptive statistics and correlation coefficients.**

|  | 1 | 2 | 3 | 4 | 5 | 6 | 7 | *M* | SD |
|---|---|---|---|---|---|---|---|---|---|
| **Survey 1** | | | | | | | | | |
| 1. Possession-self link | — | | | | | | | 2.89 | 0.81 |
| 2. Feeling of ownership | 0.39** | — | | | | | | 3.88 | 0.96 |
| 3. Perceived control | 0.21** | 0.62** | — | | | | | 5.13 | 1.32 |
| 4. WTA index | −0.03 | 0.00 | 0.01 | — | | | | 75.63 | 996.83 |
| 5. WTP index | 0.07 | −0.02 | −0.03 | 0.02 | — | | | 2.47 | 11.47 |
| 6. Endowment effect index | 0.02 | 0.00 | −0.01 | 0.09** | −0.01 | — | | 2,292.65 | 45,008.65 |
| **Survey 2** | | | | | | | | | |
| 1. Possession-self link | — | | | | | | | 3.02 | 0.85 |
| 2. Feeling of ownership | 0.37** | — | | | | | | 3.94 | 0.91 |
| 3. Perceived control | 0.24** | 0.48** | — | | | | | 5.25 | 1.20 |
| 4. WTA index | 0.07* | 0.05 | 0.08* | — | | | | 2.80 | 2.65 |
| 5. WTP index | 0.06 | 0.04 | 0.07 | 0.46** | | | | 1.51 | 1.80 |
| 6. Endowment effect index | 0.00 | −0.04 | −0.04 | 0.05 | −0.06 | — | — | 4.33 | 35.00 |
| 7. Self-extension tendency | 0.63** | 0.28** | 0.27** | 0.13** | 0.09** | 0.06 | — | 2.98 | 0.84 |

**Notes:**
WTA index: WTA/market price; WTP index: WTP/market price; endowment effect index: WTA/WTP.
WTA: willingness to accept; WTP: willingness to pay.
* $p < 0.05$.
** $p < 0.01$.

score in a positive direction ($\beta = 0.62$, $p < 0.001$). While the associations between perceived control and each factor followed our predictions for Survey 1, almost no associations were found between monetary value indexes and the two factors, countering our predictions. It may be that the value restriction of "around 30,000 yen" placed on the objects participants selected in Survey 1 resulted in biased judgments of monetary value and impacted variation in monetary value indexes. For this reason, no monetary restrictions were established for Survey 2. Self-extension tendency was also added as an indicator to examine validity.

For Survey 2, we performed multiple regression analysis with the possession-self-link score as the dependent variable and perceived control, WTA index, WTP index, endowment effect index, and self-extension tendency (Cronbach's $\alpha = 0.91$) as independent variables. The results demonstrated that only perceived control and self-extension tendency significantly predicted the possession-self-link score in a positive direction ($\beta = 0.07$, $p = 0.01$; $\beta = 0.61$, $p < 0.001$). Next, when the same multiple regression analysis was performed using the feeling of ownership score as the dependent variable, only perceived control and self-extension tendency significantly predicted feelings of ownership score in a positive direction ($\beta = 0.43$, $p < 0.001$; $\beta = 0.17$, $p < 0.001$). Thus, the associations between perceived control and self-extension tendency with each factor followed our predictions. In particular, there was a weak correlation between self-extension tendency and feeling of ownership. However, similar to Survey 1, no associations were found between monetary value indexes and either factor, countering our predictions.

### Reliability

For Survey 1, the Cronbach's alpha of POS-J overall, possession-self link and feeling of ownership were 0.86, 0.86 and 0.85, respectively. For Survey 3, intraclass correlations between administration one and administration two confirmed a sufficient level of test–retest reliability (POS-J overall: ICC = 0.63, 95% CI [0.50–0.73]; possession-self link: ICC = 0.69, 95% CI [0.57–0.78]; feeling of ownership: ICC = 0.68, 95% CI [0.56–0.77]).

## Discussion

The purpose of Study 1 was to create a POS-J. The results confirmed a high level of reliability and a two-factor structure consistent with that of previous research (*Walasek, Matthews & Rakow, 2015*). Not only did perceived control show a positive association with each factor, but the association with the feeling of ownership was also consistently stronger across both surveys, which aligned with our predictions based on previous research (*e.g.*, *Peck, Barger & Webb, 2013*). Furthermore, positive associations were observed between self-extension tendency and each factor, and the association with the possession-self link was found to be stronger, which are also consistent with our predictions based on previous studies (*e.g.*, *Ferraro, Escalas & Bettman, 2011*; *Walasek, Rakow & Matthews, 2017*). Thus, these results generally supported the validity of the POS-J.

On the other hand, although it has been shown that people estimate a higher monetary value for objects that they feel psychological ownership over (*e.g.*, *Shu & Peck, 2011*), the WTA, WTP, and endowment effect indices had no or very minimal associations with the two factors of the POS-J in Surveys 1 and 2. While we discuss the reasons for this in detail in the General Discussion, further careful study is necessary.

## STUDY 2

In Study 1, we created the POS-J and confirmed that it was highly reliable and generally valid. However, Study 1 measured psychological ownership over the participants' own possessions. In reality, psychological ownership arises for both owned and non-owned objects (*e.g.*, *Reb & Connolly, 2007*). Furthermore, psychological ownership over non-owned objects has a significant impact on purchase decisions and has been the focus of recent marketing and consumer behavior studies (*e.g.*, *Jussila et al., 2015*). Therefore, we chose to address the psychological ownership of non-owned objects in Study 2.

To do so, we focused on how haptic imagery increases psychological ownership. Haptic imagery refers to imagining touching and holding an object in one's hands and thinking about how it would feel. Prior studies have repeatedly shown that haptic imagery increases psychological ownership (*e.g.*, *Iseki & Kitagami, 2016*; *Iseki & Kitagami, 2017*; *Peck, Barger & Webb, 2013*). Therefore, in Study 2, we aimed to examine the construct validity of the POS-J by using it to retest the effect of haptic imagery (*Peck, Barger & Webb, 2013*) for non-owned objects.

## Materials and Methods

### Ethics statement

This protocol, along with that of Survey 1 of Study 1, was approved (approval number: NUPSY-200503-I-01). Study 2 was the on-site survey, thus we obtained written informed consent prior to participation.

### Participants and design

Study 2 was a one-factor (haptic imagery/no-haptic imagery) between-participants design. Participants were 253 university students in Japan (93 males, 160 females; mean age 18.57 years, SD = 1.07). Detailed information about the estimation of the necessary sample size and data exclusion criteria can be found in the Supplemental Article.

### Materials

Similar to previous studies (*e.g.*, *Iseki & Kitagami, 2016*; *Peck, Barger & Webb, 2013*), the stimulus was a picture of a blanket.

### Measurements

We rephrased the scale used in Study 1 to create a version of the POS-J (no legal ownership version) that would not be confusing when measuring the psychological ownership of non-owned objects. Specifically, with reference to previous research (*e.g.*, *Fuchs, Prandelli & Schreier, 2010*; *Kirk, Peck & Swain, 2018*), we added expressions such as "I feel as if" (*maru de* and *ka no you ni kanjiru*) to the beginning of the sentences (Table S4).

### Procedures

The experiment was conducted as part of a university class. Participants were given paper questionnaires and instructed to turn the pages only when the researcher told them to do so. They were randomly assigned to either a haptic imagery condition or a no-imagery condition to keep the number of participants as close to equal as possible.

The haptic imagery manipulation followed the method used by *Peck, Barger & Webb (2013)* and *Iseki & Kitagami (2016)*. The picture of the blanket was located on the paper questionnaires, and the word "blanket" was written above it. No additional information about the blanket was presented.

Participants in the haptic imagery condition first looked at the picture of the blanket for 30 s. They were then instructed to evaluate the blanket as if they were considering buying it while imagining "how it would feel to touch or hold the blanket from the previous page in your hands" for 1 min with their eyes closed. Meanwhile, participants in the no-imagery condition also looked at the product image for 30 s then were instructed to evaluate the product from the previous page for 1 min as if they were considering buying it. Next, participants were asked to respond to the nine items of the POS-J concerning the blanket using a five-point scale from "*completely disagree*" to "*completely agree*." We used four different patterns of item order, and the order assigned to participants was randomly determined.

**Table 4 Descriptive statistics for each condition.**

| | Haptic imagery | | No imagery | |
|---|---|---|---|---|
| | *M* | SD | *M* | SD |
| Psychological ownership | 1.86 | 0.72 | 1.50 | 0.63 |
| Possession-self link | 1.63 | 0.67 | 1.43 | 0.62 |
| Feeling of ownership | 2.33 | 1.11 | 1.64 | 0.82 |

We did not account for actual ownership status, as psychological ownership is exhibited regardless of whether one owns a similar object or not (this has been described in many previous studies, *e.g.*, *Peck & Shu, 2009*).

### Data analysis

Levene's test was performed to explore the homogeneity of variance. Subsequently, we conducted a *t*-test or a Welch's *t*-test with haptic imagery as the independent variable and the psychological ownership score and subscales (*i.e.*, possession-self-link, feeling of ownership) as the dependent variable.

## Results

In this study, all analyses were performed using the R version 4.1.2 (R Development Core Team, Vienna, Austria) software package.

Data from 252 participants (93 males, 159 females; mean age 18.57 years, SD = 1.07) were analyzed, excluding one student with missing responses to the POS-J questions. There were 127 participants (49 males, 78 females) in the haptic imagery condition and 126 participants (44 males, 82 females) in the no-imagery condition.

We averaged the rating scores of items of the POS-J (9 items; Cronbach's $\alpha$ = 0.89) to calculate the psychological ownership score. As psychological ownership has a two-factor structure, we also averaged ratings to calculate the possession-self-link score (6 items; Cronbach's $\alpha$ = 0.86) and feeling of ownership score (3 items; Cronbach's $\alpha$ = 0.87). The mean and standard deviation for each score for each condition is shown in Table 4.

The results of Levene's test confirmed homogeneity of variance for psychological ownership score and possession-self link score ($p$ = 0.13; $p$ = 0.38), but not for feeling of ownership score ($p < 0.001$). Therefore, we performed a Welch's *t-test* for the feeling of ownership score. A *t*-test with haptic imagery as the independent variable and psychological ownership score as the dependent variable was significant ($t$ (250) = 4.26, $p < 0.001$, Cohen's $d$ = 0.54), as was a *t*-test with haptic imagery as the independent variable and possession-self-link score as the dependent variable ($t$ (250) = 2.43, $p$ = 0.02, Cohen's $d$ = 0.31). A Welch's *t*-test with haptic imagery as the independent variable and feeling of ownership score as the dependent variable was also significant ($t$ (231.54) = 5.65, $p < 0.001$, Cohen's $d$ = 0.71).

## Discussion

The purpose of Study 2 was to examine construct validity by using the POS-J to retest previously conducted methods used in research of non-owned objects (*e.g.*, *Peck, Barger &*

*Webb, 2013*). Scores for psychological ownership, possession-self link, and feeling of ownership were all higher in the haptic imagery condition than in the no-imagery condition, supporting the construct validity of the POS-J.

## GENERAL DISCUSSION

The purpose of this study was to develop a POS-J. The results of Study 1 demonstrated that the POS-J has the same two-factor structure (possession-self link and feeling of ownership) found in *Walasek, Matthews & Rakow (2015)* and satisfactory goodness of fit. Concerning reliability, we examined the reliability of POS-J subscales to estimate Cronbach's alpha coefficient as a parameter of internal consistency, and intra-class correlation coefficient as a parameter of test–retest reliability. The results showed that Cronbach's alpha coefficient (POS-J overall: $\alpha = 0.86$; possession-self link: $\alpha = 0.86$; feeling of ownership: $\alpha = 0.85$, in Survey 1) and intra-class correlation coefficient (POS-J overall: ICC = 0.63; possession-self link: ICC = 0.69; feeling of ownership: ICC = 0.68, in Survey 3) were at acceptable levels. Therefore, we determined POS-J had sufficient reliability. With respect to validity, our results were also largely consistent with our predictions. Specifically, in Study 1, perceived control was positively associated with each factor, and the association with the feeling of ownership was stronger across Surveys 1 and 2. Furthermore, the self-extension tendency was also positively associated with each factor, and the association with the possession-self link was strong; nevertheless, the association with the feeling of ownership was weak. These results, which were consistent with our predictions based on previous studies (*e.g.*, *Ferraro, Escalas & Bettman, 2011*; *Peck, Barger & Webb, 2013*; *Walasek, Rakow & Matthews, 2017*), are believed to provide a sufficient level of support for the criterion related validity and the construct validity of the POS-J.

In this research, we also aimed to create a version of the scale that could measure the psychological ownership of non-owned objects (*i.e.*, no legal ownership version). Accordingly, in Study 2, we focused on the effects of "haptic imagery" (*e.g.*, *Peck, Barger & Webb, 2013*) and examined whether viewing a picture of a non-owned object and imagining touching it would increase psychological ownership. To do so, we rephrased the questions on the scale. The results confirmed the effect of haptic imagery on the possession-self link and the feeling of ownership. The effect size was greater for the feeling of ownership (Cohen's $d = 0.71$) than for the possession-self link (Cohen's $d = 0.31$). Previous research has shown that haptic imagery promotes perceived control, resulting in increased feelings of ownership (*Iseki & Kitagami, 2016*; *Peck, Barger & Webb, 2013*). Therefore, Study 2, which found that the effect size of haptic imagery was greater for the feeling of ownership (also consistent with previous studies), supported the scale's validity. The only stimulus used in Study 2 was a blanket. As the effects of haptic imagery have already been confirmed with a variety of stimuli (*e.g.*, *Iseki & Kitagami, 2016*; *Iseki & Kitagami, 2017*; *Peck, Barger & Webb, 2013*), there is a high probability that similar results could be obtained using stimuli other than a blanket, but it would be optimal to confirm this before conducting a preliminary study using this instrument to measure the psychological ownership of non-owned objects.

Research on psychological ownership covers a wide range of topics, from basic research in cognitive psychology to applied studies in organizational psychology and consumer behavior. Nonetheless, the scales used to measure psychological ownership were not consistent. Moreover, each study has had its own way of measuring the psychological ownership of non-owned objects, such as by rephrasing scale items (*e.g.*, *Fuchs, Prandelli & Schreier, 2010*; *Kirk, Peck & Swain, 2018*). Thus, it is difficult to compare and discuss across studies because they have not used a consistent scale. Further, it has been suggested that the mechanism by which psychological ownership arises differs between the West and Japan (or Asia) for various reasons, including differences in the tendency to associate control with ownership (*Pierce, Kostova & Dirks, 2003*). It is hoped that the development of the POS-J presented in this study will further clarify the mechanisms of psychological ownership by considering cultural differences through further studies on psychological ownership among studies of Japanese populations and in cross-cultural research.

The present study failed to replicate significant associations between monetary value indicators (WTA index, WTP index, endowment effect index) and either factor in Survey 1 or Survey 2 of Study 1, despite previous research (*e.g.*, *Shu & Peck, 2011*) showing that people estimate a high monetary value for objects for which they feel psychological ownership. This difference might stem from variations in the methods used. In many classical studies of the endowment effect, participants providing WTA responses about objects they actually owned, while participants providing WTP responses did not (*e.g.*, *Kahneman, Knetsch & Thaler, 1990*). On the other hand, Study 1 followed the procedure of *Walasek, Matthews & Rakow (2015)* and measured WTP by asking participants to call to mind a cherished possession and suppose that they lost it. Thus, the WTP measured in Study 1 may have deviated from the "pure" WTP for non-owned objects addressed in classical endowment effect studies. *Morewedge et al. (2009)* found that when a participant providing a WTP response also owns the object, they estimate a higher WTP; thus, there is no gap with WTA, and no endowment effect is observed. This explanation may account for our inability to measure WTP and the endowment effect in Study 1 accurately.

The failure to replicate significant associations between monetary value indicators might also be due to the influence of cultural differences. For example, the gap between WTP and WTA is larger for people in Western cultures than those in Eastern cultures (*e.g.*, *Maddux et al., 2010*). Furthermore, it has been found that the endowment effect is greater in Japanese populations when the association between self and object is weak rather than strong, but the opposite pattern has been seen with Canadians of European descent (*Maddux et al., 2010*). It has been suggested that this difference is due to self-criticism among the Japanese (*Heine & Hamamura, 2007*; *Heine et al., 1999*; *Kitayama et al., 1997*). In other words, the strong tendency for self-criticism (as opposed to self-enhancement) leads to a low evaluation of objects that are strongly linked to the self (*Maddux et al., 2010*), which would suggest that the endowment effect was smaller in Study 1 because participants were asked to think of an object that was important to them and had a strong link to their sense of self.

Furthermore, collectivism among Japanese may also influence the results, as collectivism has been negatively associated with psychological ownership (*Menard,*

*Warkentin & Lowry, 2018*). Thus, it may be that the emergence of psychological ownership was suppressed in this study, which was conducted in Japanese, and this influenced monetary value indexes. In fact, *Iseki & Kitagami (2016)* in Japan also found no association between psychological ownership and monetary value. Future research may further discuss the influence of collectivism and tendencies toward self-criticism by using the POS-J to compare cultures.

This study also has the following limitation. There might be potential biases in the selection of participants. Since we recruited participants through crowdsourcing services in Study 1, there may have been biases in the backgrounds of the participants, such as income, work, and education. In addition, we did not control for participants' nationality, although almost all of them are considered Japanese. At least, the participants understand the Japanese language sufficiently to use the crowdsourcing service and understand the instructions in the survey. This limitation possibly affects the generalizability of our findings. The failure to replicate significant associations between monetary value indicators as mentioned above may have been due to this sampling bias. It is important to conduct additional studies, where various demographic factors are considered.

Lastly, there is also a practical trend of attempting to accurately understand the psychological ownership we feel in various contexts and use it in management and marketing. For example, amid the rise of various services that seek to shift consumers from ownership to *use* or to *sharing* (sharing economies), it is increasingly important to gain an accurate understanding of psychological ownership people feel over non-owned objects. Due to the ability to assess both aspects of psychological ownership, the POS-J can be used to explore the psychological ownership in both traditional ownership and sharing, as well as the differences therein. Thus, the POS-J can also contribute to psychological ownership research through the collaboration of industry and academia.

## CONCLUSIONS

This study examined the factor structure, reliability, and validity of the POS-J, a new instrument for use in Japan. The item-factor structure of the POS-J was identical to that of its original English version (*Walasek, Matthews & Rakow, 2015*). The POS-J comprises two subscales: possession-self link and feeling of ownership. Both subscales had adequate internal consistency and test–retest reliability. The results of this study demonstrate the construct validity of the POS-J. Although there were some limitations, the present study provides the first empirical support for the Japanese version of the two subscales POS-J.

### Funding

This research was supported by JSPS KAKENHI: JP20K22274 to Sayo Iseki, and JP17J05236 and JP19K14482 to Kyoshiro Sasaki. The funders had no role in study design, data collection and analysis, decision to publish, or preparation of the manuscript.

## Grant Disclosures
The following grant information was disclosed by the authors:
JSPS KAKENHI: JP20K22274, JP17J05236, and JP19K14482.

## Competing Interests
The authors declare that they have no competing interests.

## Author Contributions
- Sayo Iseki conceived and designed the experiments, performed the experiments, analyzed the data, prepared figures and/or tables, authored or reviewed drafts of the paper, and approved the final draft.
- Kyoshiro Sasaki conceived and designed the experiments, performed the experiments, analyzed the data, prepared figures and/or tables, authored or reviewed drafts of the paper, and approved the final draft.
- Shinji Kitagami conceived and designed the experiments, performed the experiments, authored or reviewed drafts of the paper, and approved the final draft.

## Human Ethics
The following information was supplied relating to ethical approvals (*i.e.*, approving body and any reference numbers):

This study was approved by the Ethics Committee on Human Experimentation at Nagoya University, Japan (approval number: NUPSY-200503-I-01) and Chukyo University, Japan (approval number: 2020-001).

## Data Availability
The raw data and the POS-J are available in the Supplemental File.

## Supplemental Information
Supplemental information for this article can be found online at http://dx.doi.org/10.7717/peerj.13063#supplemental-information.

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
