# Peer review of "Development of a Japanese version of the Psychological Ownership Scale"

_PeerJ, doi:10.7717/peerj.13063_

## Round 0.1 · original submission · Minor Revisions

Thank you for submitting your manuscript to PeerJ. I have now received two reviews, and I would like to thank both reviewers for their thoughtful feedback on the paper. Both reviewers see merit in the manuscript; however, both also have a number of comments that require addressing. The reviews are appended below, so I will not reiterate all the comments here.

However, I did also have a few additional comments that I believe would strengthen the manuscript.

1) In the PCA in Study 1, you used an oblique rotation (oblimin), I assume because you expected the components to be correlated? Why then in the EFA did you use an orthogonal rotation (varimax)?

2) Related to the point above, in the CFA did you specify the factors to be correlated? This needs to be clarified, and the decision for correlating (or not) should be justified.

3) I think it would be useful to present the factor loadings (and inter-factor correlations, if specified) from the CFA in the manuscript.

Thank you for again submitting your manuscript to PeerJ and I hope you find these comments, along with the reviews, useful in revising your manuscript.

·

Basic reporting

The structure of the article is required further improvement.

Experimental design

Research question is meaningful, and methods are sufficient.

Validity of the findings

Conclusions were well stated, however, limitations were not well written.

Additional comments

Thank you for the opportunity to review your manuscript.
Authors conducted development and validation of the very important questionnaire.
I would appreciate some clarification on following issues.

Many contents are described in different sections.
Ex. Study procedure or ethical comments were written in discussion. You need to note the things about material and methods in the ‘Methods’ instead of ‘Discussion’.

Introduction
Line 165-185
These contents should be included in the method.

Material and Methods
It was not described statistical analyses at all like statistical soft, statistical criteria. The criteria to judge the validity and reliability as satisfied also should be noted here.

Results
You need to explain other demographic data like Income, job, and education including missing value. You used the crowdsourcing service, including people have various background.

Line 292-322
Basically, statistical analyses conducted in the research are required to explain in the method prior to results. Many statistical words are found firstly found in Results, not introducing in Methods such as KMO, Bartlett, Cronbach’s α, AGFI, and so on.

P294-296
The following description is not ‘Results’, but ‘Methods’.
the usage of principal components analysis

Line 306-308
The following description is not ‘Results’, but ‘Methods’.
Exploratory factor analysis, KMO, and Bartlett’s test

Discussion
Line 358-542
In discussion section, following contents are included.
materials and methods, participants and design, materials, measurements, procedures, results, discussion, general discussion.
This made readers confusing. Ex. Material and methods section should be material methods section not discussion section. Thorough amendments should be needed to be easily to read.
You should focus on the content of discussion and conclusion.

Line 325-351 Validity testing
You firstly need to explain about how to prove the validity and reliability testing of your questionnaire in the Methods.
For example, validity testing contains content validity, criterion/criterion-referenced validity, construct validity such as convergent validity and discriminant validity.

Yang, H., Shi, L., Lebrun, L. A., Zhou, X., Liu, J., & Wang, H. (2013). Development of the Chinese primary care assessment tool: data quality and measurement properties. International journal for quality in health care, 25(1), 92-105.
Dullie, Luckson, et al. "Development and validation of a Malawian version of the primary care assessment tool." BMC family practice 19.1 (2018): 1-11.

Line 353 Test-retest reliability
Which correlations analysis was selected in these analyses?
You should also explain in the Methods.

Line 359
You need to say how you prove the high level of reliability

Line 374-391
These contents are mostly ‘Methods’ or ‘Results’.

Line 394-397
This section is discussion. Mistake? Why do you describe here?
In addition, ethical contents were already described in the previous ‘Methods’ section, too.

Line 400-406
These are mostly ‘Results’ contents like participants, missing value. Sample size and exclusion criteria are ‘Methods’.

Line 409-412
These are ‘Methods’ since this is a procedure you adopted how to measure.

Line 415-419
This needs to be noted in the ‘Results’.

Line 421-434
Some descriptions are in the ‘Methods’ like how participants look and evaluate the blanket for the hepatic imagery and how to evaluate, others are in the ‘Results’ like the number of participants.

Line 437-451
These descriptions should be noted in the ‘Results’. Based on the ‘Results’, you need to discuss.

Line500-505
Please discuss the failure which your results are different from previous studies more. For example, participants’ recruitment.

Limitation
Authors should also describe about the limitation such as selection biases and generalizability.
Ex. Authors conducted this study using crowdsourcing service.

Reviewer 2 ·

Basic reporting

In this article, the authors developed and validated a Japanese version of the well-established Psychological Ownership Scale in two studies focusing on owned and non-owned objects, respectively.
In my opinion, the article is very well-written, and the quality of the English is excellent. Tables are clear and raw data shared.

I have some remarks/suggestions for the introduction:

Line 41: I'm not convinced that psychological ownership is exclusively recognized by the individual and not by society, please clarify

Lines 66-68: Is there any causal relationship between the self and owned object evaluation?. I see that correlational evidence is reported. Please clarify

Lines 73-76: The superior evaluation of owned vs. others object is modulated by the evaluation of oneself and cultural factors? please clarify

Line 89-911: The authors cite the study by Peck & Schu (2009) on the role of touch in object ownership. In my opinion, it would be helpful to provide a brief explanation of why this is the case.

Section: Purpose of the study:

In my opinion, this section should start by stating the purposes of the study. I see that the authors report them starting from line 151. Please consider restructuring this section

Experimental design

The research question is clear, and the authors provide a solid background for the need to develop a validated instrument in the Japanese population

The methodology and the statistical approach sound, and the results are reported in detail and accurately.

I have one suggestion. Please report the nationality of participants (although I assume they were all Japanese?)

Validity of the findings

The authors performed confirmatory and controlled analyses, showing the validity of their results.
About the impact, this article may be useful for subsequent studies conducted in Japan as a validated scale is here provided

Additional comments

no comments

---

## Round 0.2 · accepted · Accept

Thank you for addressing the reviewer comments. I am delighted to accept your paper for publication in PeerJ.

I do have one small textual edit to recommend. I am happy for the footnote to indicate that the rotation for the EFA was changed at my recommendation; however, I think the footnote should also reflect the rationale for this (i.e. that, consistent with the PCA, you expected the factors to be correlated - hence the use of an oblique rotation).